# Improving RT-LAMP detection of SARS-CoV-2 RNA through primer set selection and combination

Yinhua Zhang, Nathan A. Tanner[ID]*

Research Department, New England Biolabs, Ipswich, Massachusetts, United States of America

* tanner@neb.com

**Data Availability Statement:** All relevant data are within the manuscript and its Supporting Information files.

**Funding:** The authors received no specific funding for this work.

## Abstract

Reverse transcription loop-mediated isothermal amplification (RT-LAMP) has emerged as a viable molecular diagnostic method to expand the breadth and reach of nucleic acid testing, particularly for SARS-CoV-2 detection and surveillance. While rapidly growing in prominence, RT-LAMP remains a relatively new method compared to the standard RT-qPCR, and contribution to our body of knowledge on designing LAMP primer sets and assays can have significant impact on its utility and adoption. Here we select and evaluate 18 LAMP primer sets for SARS-CoV-2 previously identified as sensitive ones under various conditions, comparing their speed and sensitivity with two LAMP formulations each with 2 reaction temperatures. We find that both LAMP formulations have some effects on the speed and detection sensitivity and identify several primer sets with similar high sensitivity for different SARS-CoV-2 gene targets. Significantly we observe a consistent sensitivity enhancement by combining primer sets for different targets, confirming and building on earlier work to create a simple, general approach to building better and more sensitive RT-LAMP assays.

## Introduction

The ongoing COVID-19 pandemic has brought an urgent demand for molecular diagnostic testing at an unprecedented scale. Reverse transcription quantitative PCR (RT-qPCR) has long been the standard for molecular testing and has been widely and massively used for SARS-CoV-2 detection. However, due to the large volume of testing required worldwide (e.g. 1–2 million daily tests in the US alone) and the need for testing outside of clinical laboratories, molecular diagnostic methods other than RT-qPCR have also become widely used. Isothermal amplification methods such as transcription mediated amplification (TMA) and nicking enzyme amplification reaction (NEAR) are the core chemistry for test platforms from Hologic and Abbott, respectively, with other methods showing promise for potential widespread testing. Of these alternatives, reverse transcription loop-mediated isothermal amplification (RT-LAMP) has now been used most prominently [1, 2] with numerous diagnostic tests based on RT-LAMP receiving Emergency Use Authorization from the FDA including the first ever at-home, over-the-counter molecular diagnostic test from Lucira Health [3–6].

**Competing interests:** New England Biolabs (www.neb.com) has funded this study. All authors are employees and shareholders of New England Biolabs, manufacturer of LAMP reagents described in the manuscript. This does not alter our adherence to PLOS ONE policies on sharing data and materials.

The widespread interest in RT-LAMP for molecular detection of SARS-CoV-2 has generated a lot of interest and information on how to achieve efficient and reliable detection. LAMP is a much newer method than PCR, with protocols and assay design methods not quite as established, though the recent increase in LAMP development efforts is rapidly changing this discrepancy. Many factors can have an impact on LAMP assay performance: sample source (nasopharyngeal or nasal swab, saliva); sample processing (direct sample or purified RNA); and uniquely for LAMP the detection modality and instrument/device design, e.g. pH-based colorimetric, fluorescence, Cas enzyme cleavage, etc. And regardless of test design, a critical factor to performance is of the assay primers and reaction conditions for sensitive RNA detection.

Typical RT-LAMP reactions use a primer set covering 8 regions on the target sequence and the primer design is facilitated by software (Primer Explorer V5 at Eiken https://primerexplorer.jp/e/, or LAMP Primer Design Tool at NEB https://lamp.neb.com/#!/) based on oligo length, AT and GC content and thermodynamic stability. However, like primers for RT-qPCR, not all software-designed primers perform optimally and it is often necessary to screen several sets to obtain ones that give satisfactory specificity and sensitivity. Among the published works evaluating RT-LAMP for SARS-CoV-2 detection, most screened many sets of primers before deciding on one set to continue their study. For example, 35 sets were screened by Yang et al [7] to obtain 3 sets targeting 3 different genomic regions for conventional RT-LAMP reaction; 29 sets were screened by Joung et al [6] to a single set that worked well in a coupled one-pot RT-LAMP/CRISPR cleavage assay.

To date, many sets of SARS-CoV-2 LAMP primers have been published and some used in EUA or CE-IVD diagnostic tests. Comparison of these assays and primer sets based on published data can be challenging, as methods, reagents, template sources, and other differences may have effects on sensitivity and complicate data interpretation. Here we describe evaluating RT-LAMP primer sets under identical conditions for a fairer comparison of that critical reaction parameter. Similar analyses performed previously have suggested a wide range of sensitivity among published primer sets [8, 9]. Building on these studies we apply here a more stringent approach, performing large numbers of repeats and evaluating performance with different RT-LAMP reagents and amplification temperatures, running >5,000 RT-LAMP reactions to more fully evaluate the performance of SARS-CoV-2 assays. Significantly, we demonstrate a general principle to further increase LAMP detection sensitivity by combining primer sets in the same reactions, expanding on our previous observations [10].

## Materials and methods

### RT-LAMP primer selection

17 LAMP primer sets from previous publications and 1 new primer set were selected for evaluation (S1 Table). 3 sets (N2, E1 and Ase1) were studied in our previous publication [10]. 10 sets (S2 [11], S4 [12], S10 [12], S11 [13], S12 [3], S13 [7], S14 [14], S17 [12], S18 [15], Mam-N/S19 [3]) were shown to be the most sensitive ones in a previous analysis of 19 published sets [8]. 5 primer sets (S-Huang [16], N-Baek [17], As1e [18], S-Yan [13], N-Lu [15]) were screened as the most sensitive sets out of 16 published sets [9]. These included 2 sets (S-Yan = S11, N-Lu = S18) that overlapped with those selected from Dong et al. A new set (SGF-wt) was selected based on sensitive detection at a region of SARS-CoV-2 variant sequence deletion in our testing. Primer set Joung was from Joung et al [6] and Lau from Lau et al [19]. All primer sequences are shown in S1 Table.

All primers were synthesized by IDT at 100 or 250 nmol scale with standard desalting. Primers were dissolved and then mixed in ddH2O as 25x stocks of each set based on standard

1x final concentrations in LAMP: 0.2 μM F3, 0.2 μM B3, 1.6 μM FIP, 1.6 μM BIP, 0.4 μM Loop F, 0.4 μM Loop B.

### RT-LAMP reactions

RT-LAMP reactions were performed using either WarmStart® Colorimetric LAMP 2X Master Mix (DNA & RNA) (M1800) or WarmStart® LAMP Kit (DNA & RNA) (E1700) containing *Bst* 2.0 WarmStart and WarmStart RTx polymerases from New England Biolabs (NEB). 40 mM guanidine hydrochloride was included in all reactions to improve LAMP reaction speed and sensitivity [10]. The same vial of synthetic SARS-CoV-2 RNA from Twist Bioscience (Twist Synthetic SARS-CoV-2 RNA Control 2 (MN908947.3), SKU: 102024) was used for all reactions, with SARS-CoV-2 RNA diluted and aliquoted in nuclease-free water and 10 ng/μl Jurkat total RNA (Biochain). RT-LAMP reactions were performed in 25 μl volumes with 1 ul of diluted SARS-CoV-2 RNA supplemented with 1 μM SYTO®-9 double-stranded DNA binding dye (Thermo Fisher S34854) in 96-well plates and incubated at 65 or 60˚C on a real-time qPCR machine (BioRad CFX96). Amplification signal was acquired every 15 seconds for 108 "cycles" (total incubation time was ~40 min). The time to reach the signal threshold (Tt) was determined from the real time fluorescence signal and positive was scored using an arbitrary cutoff of 22.5 minutes for RT-LAMP at 65˚C and 30 minutes at 60˚C in order to compare LAMP efficiency. For each primer set or combination test condition, a minimum of 24 repeats were performed, and the number of positives were used as an indicator for detection sensitivity. At least 8 repeats of no-template control reactions were performed for each primer set or combination to evaluate production of target-independent signal. A low RNA template amount was used to ensure only a percentage of positive reactions in the repeats, but not in all in order to compare relative performance. To ensure a fair comparison and consistent reagent performance we used the same lots of reagents for all tests and included 24 repeats of N2 primer set with 50 copies of SARS-CoV-2 RNA at each assay date and confirmed consistent reaction times and positive detection results.

## Results

### Procedure to identify sensitive primer sets

RT-LAMP reactions were tested with both M1800 (pH-based colorimetric) and standard E1700 LAMP mixes. Both formulations were tested at 65˚C and 60˚C to capture any temperature preference by primers. We first tested single primer sets using 50 copies of SARS-CoV-2 synthetic RNA in each 25 ul reaction, with further evaluation of the most sensitive primer sets in various combinations with 25 and 12.5 copies of RNA template. The sensitivity of RT-LAMP was evaluated based on the number of positives in 24 repeats of reactions each containing 50 copies of SARS-CoV-2 RNA (Fig 1), which is at the limit of detection of most sensitive primer sets. Each primer set was classified as sensitive, medium or poor based on the number of positives in the repeats. With M1800 colorimetric LAMP at 65˚C (Fig 1A), 8 primer sets (S4, S10, S11, S12, S13, S18, N2 and E1) were classified as sensitive, 7 medium (S2, S14, S17, Joung, As1e, SGF-wt and Mam-N) and 3 poor (N-Baek, S-Huang and N-Lau). With M1800 at 60˚C (Fig 1B), 7 sets (S10, S11, S12, S13, S17, N2 and E1) were classified as sensitive. 6 of those 7 sets were consistent at the two temperatures, but S4 and S18 showed high sensitivity at 65˚C that was reduced at 60˚C, and one set, S17, showed increased sensitivity at 60˚C.

RT-LAMP speed, indicated by the average Tt of positives, was generally consistent across the 18 primer sets, ranging from 10.8–17.8 minutes with M1800 and 50 copies at 65˚C. There was some association with the faster primer sets and increased sensitivity, but correlation of positive detection with Tt was very weak ($R^2 = 0.23$) implying that the two observables are not

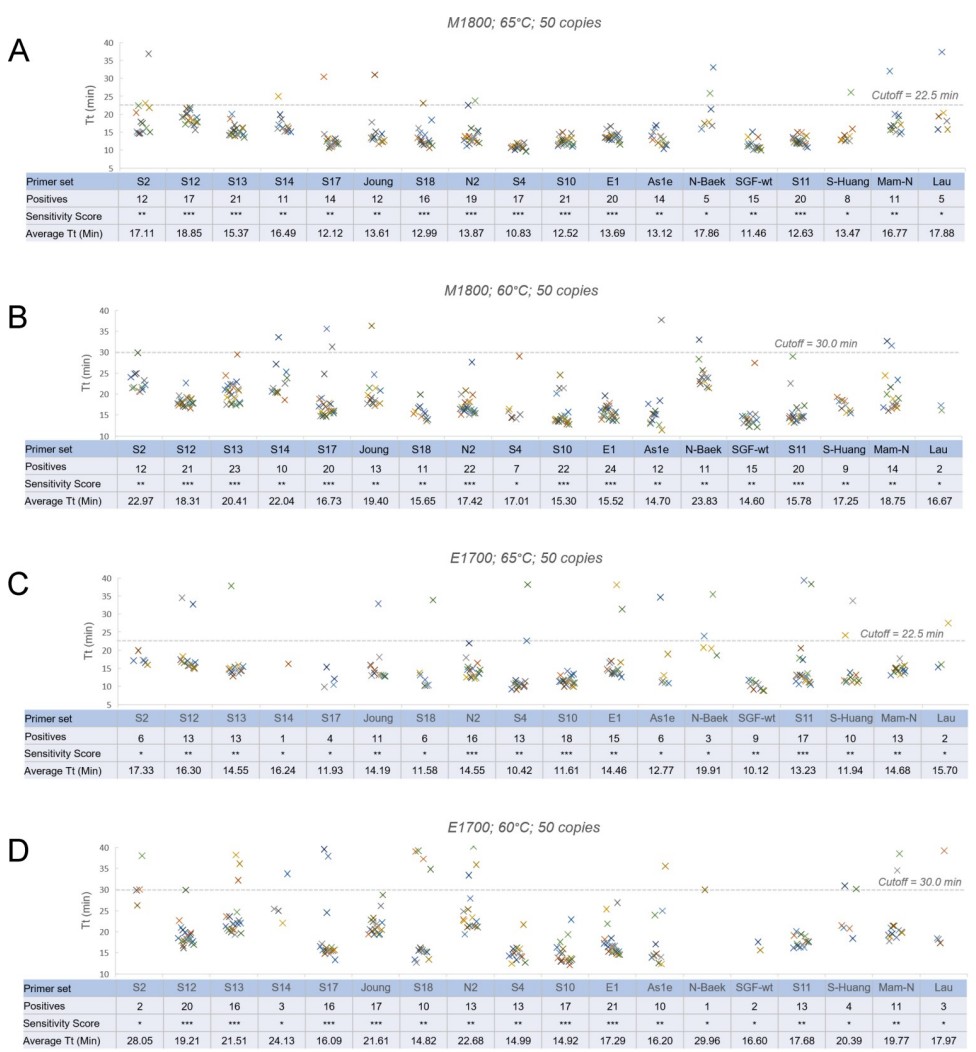

**Fig 1. Performance of single primer sets.** Each primer set was assayed with 24 repeats with 50 copies of SARS-CoV-2 RNA template per reaction. Each primer set was assigned a sensitivity score: sensitive (*** 16–24 positives); medium (** 9–15 positives) or poor (* 1–8 positives). (A) RT-LAMP reactions conducted with M1800 Colorimetric LAMP at 65°C. (B) With M1800 at 60°C. (C) With E1700 LAMP Mix at 65°C. (D) With E1700 at 60°C.

necessarily linked. Reaction speed for 16 of the 18 primer sets decreased (by 3.9 ± 1.5 minutes) when temperature was reduced to 60°C, with only S12 and Lau sets increasing slightly in speed; with E1700 all sets were slower at 60°C. Taken together these results indicate that reaction speed is determined primarily by the polymerase amplification chemistry and not primer sequence.

When assayed with E1700, primer sets that gave high sensitivity with M1800 also showed higher sensitivity, however the overall positive numbers were slightly lower. There were 3 sets (S10, S11 and N2) in the sensitive category at 65°C (Fig 1C) and 6 sets (S10, S12, S13, S17, E1 and Joung) in this group at 60°C (Fig 1D). Primers that showed poor sensitivity in M1800 (N-Baek, S-Huang and N-Lau) as well as 5 additional sets (S2, S14, S17, S18 and As1e) performed poorly in E1700 at 65°C. Among them, 3 sets (S17, S18 and As1e) showed improvement at 60°C, moving to the medium category.

## Improving sensitivity by combining two sensitive primer sets

Sensitive primer sets were selected to test their performance in RT-LAMP reaction containing 2 primer sets. 5 sets (S10, S11, S12, S17, N2 and E1) were chosen based on high sensitivity using both RT-LAMP formulations and temperatures (S13 was one of the most sensitive sets but it was not selected here because its primers differ in only a few bases from primer set E1). RT-LAMP containing 2 primer sets was compared to a single primer set in the presence of 25 copies of SARS-CoV-2 RNA using M1800 at 65 and 60˚C (Fig 2A and 2B). In all the combinations tested (S10+S11; S10+S12; S11+S12; and N2+E1), the combined primer reactions gave higher sensitivity than with single primer sets at both temperatures, except S12+S17 which matched the performance of S12 at 65˚C but increased sensitivity at 60˚C. The performance of these best dual primer sets was further evaluated side by side with both RT-LAMP formulations and at both 65 and 60˚C (Fig 2C and 2D). The results showed that there were slight differences in their performance regarding formulation and incubation temperature. N2+E1 showed slightly higher sensitivity at 60˚C in both formulations while combinations of S10, S11, and S12 showed the same tendency only in E1700.

We next extended this comparison with even lower SARS-CoV-2 RNA templates (~12.5 copies) to test how these primers perform as single or dual primer sets (Fig 3). 4 primer sets were divided into two groups (S10, S11; and N2, E1) and the single primer set was compared with dual primer set from each group (S10+S11, N2+E1). As shown, combination of 2 primer sets S10+S11 (Fig 3A and 3B) was clearly more sensitive with M1800 at 60˚C (20/24 positives vs 13/24 and 8/24 single sets) but less pronounced at 65˚C (13/24 positives vs 8/24 and 6/24 single sets). A similar trend was observed with E1700 but with a lower positive frequency in all cases than with M1800. N2+E1 (Fig 3C and 3D) also showed more sensitive detection than each single primer set with M1800 at both 65˚C (16/24 positives vs 9/24 and 6/24 single sets) and 60˚C (18/24 positives vs 10/24 and 9/24 single sets) and with E1700 at 60˚C (12/24 positives vs 4/24 and 5/24 single sets) but less clear at 65˚C (7/24 positives vs 6/24 and 3/24 single sets). These results reinforced the concept that RT-LAMP with dual primer set is more sensitive even at low template and for some primer sets use of 60˚C can provide increased sensitivity albeit with potentially increased reaction times.

## Combination of 3 primer sets

As use of 2 primer sets showed better detection sensitivity than a single set, we evaluated whether adding a third set could further improve the detection sensitivity. We assayed two groups (S10, S11, S12; and N2, E1, As1e) with all pairwise combinations of primer sets within each group versus reactions containing all 3 sets. LAMP reactions were performed with ~12.5 copies of SARS-CoV-2 RNA template. Overall, the benefit of 3 primer sets over 2 sets was not consistent across different primer combinations. For S10+S11+S12 (Fig 4A and 4B), there seemed to be no advantage and it gave similar number of positives as those reactions with only 2 sets of primers combinations (S10+S11, S10+S12 and S11+S2) in both RT-LAMP reagents and at either reaction temperature. For N2+E1+As1e (Fig 4C and 4D), there was some advantage, and it produced more positives than those reactions containing 2 primer sets (N2+E1, N2 +As1e and E1+As1e) with M1800 especially at 60˚C but not with E1700. While of course it is possible that other combinations of 3 primer sets may provide a sensitivity benefit to RT-LAMP, the results seen here indicate diminishing returns adding a 3[rd] set to a combination of 2 primer sets, and that a combination of 2 sets is likely a sufficient approach to increasing sensitivity.

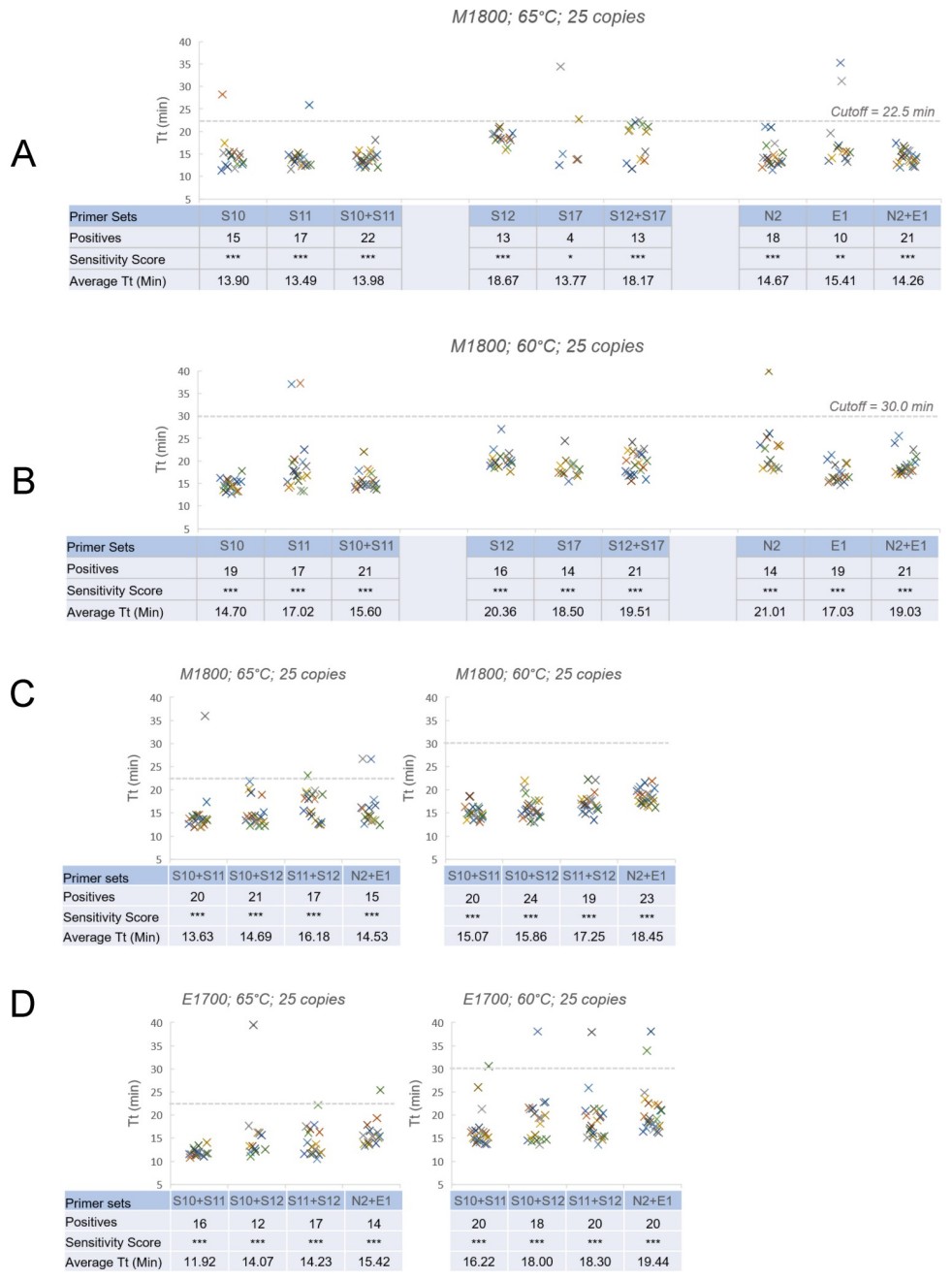

**Fig 2. Increased detection sensitivity by combining 2 primer sets.** Reactions were performed with single primer sets and various combinations of 2 primer sets in the presence of 25 copies of SARS-CoV-2 RNA template per reaction. The sensitivity score was adjusted as: sensitive (*** 12–24 positives); medium (** 6–11 positives) or poor (* 1–5 positives). (A) Reactions with single primer sets were compared to combinations of 2 sets using M1800 at 65˚C. (B) With M1800 at 60˚C. (C) Comparing reactions with 4 different dual primer set combinations (S10+S11, S10+S12, S11 +S12 and N2+E1) in M1800 at 65 and 60˚C. (D) The same dual primer sets in E1700 at 65 and 60˚C.

## Discussion

Through the evaluation of 18 SARS-CoV-2 primer sets, we have a better understanding of the relative performance of these primers. Among them, 6 primer sets (S10, S11, S12, S13, N2 and E1) gave the most sensitive SARS-CoV-2 RNA detection with two LAMP formulations and

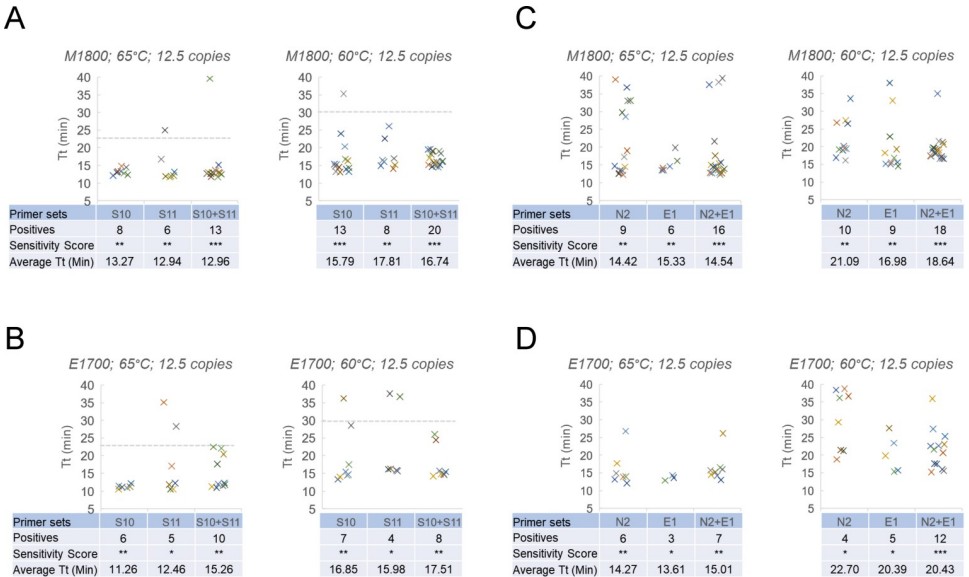

**Fig 3. Effect of combining 2 primer sets in the presence of extremely low SARS-CoV-2 RNA template.** A similar test was designed as Fig 2 but using only ~12.5 copies of SARS-CoV-2 RNA template per reaction. (A) Single primer set reactions for primers S10 and S11 versus a combination of them in M1800 at 65 and 60˚C. (B) In E1700 at 65 and 60˚C. (C) Single primer set reactions for primers N2 and E1 versus a combination of them in M1800 at 65 and 60˚C. (D) In E1700 at 65 and 60˚C. Sensitivity Score for 12.5 copies: *** 12–24 positives; ** 6–11 positives; * 1–5 positives.

temperatures. 2 primer sets (S4 and S18) showed high sensitivity with M1800 Colorimetric LAMP only at 65˚C, and 2 sets (S17, Joung) showed high sensitivity at 60˚C with both M1800 and E1700. Some of these primer sets were also shown to be among the most sensitive in previous comparisons. For example, S4, S10, S11, S13 were among the 6 final sets in Dong et al (along with S14 and S17) [8]. Their study gave more weight of scoring to primer sets that had faster reaction speed instead of percentage of positives, but we observed in comparison here minimal differences in speed across the sets and little correlation with speed and sensitivity. S11 and S18 appeared in the final 5 along with N-Baek, S-Huang and As1e in Janikova et al [9] though this work assayed only a small number of reactions in comparing the sets. The ability to design LAMP primers in provided by LAMP primer tools, but empirically testing each designed set for performance remains the best strategy to identify highest performance. Adjustments to designed primers such described by [20] can improve curve shape and specificity, but for maximum sensitivity screening and combination are the best suggestions for assay development.

Overall we did observe some consistent trends regarding to reaction formulation and temperature, with a slight improvement in sensitivity with the M1800 Colorimetric LAMP as compared to the general-purpose E1700. Optimal reaction temperature varied for the 18 sets. Using M1800, 13 sets showed similar performance at the two temperatures, 3 more sensitive at 60˚C and 2 more sensitive at 65˚C. Using E1700 there was more variation, with 6 sets giving similar results, 7 more sensitive at 60˚C and 5 more sensitive at 65˚C. In general, primers performed well and primers that are poor remains the same in both recipes and reaction temperatures, suggesting that the identification of good RT-LAMP primer is the most important step toward sensitive and reliable assay.

We further showed that the detection sensitivity could be increased by combining 2 sensitive primer sets into the same RT-LAMP reaction, for both temperatures and LAMP formulations evaluated. This strategy was described in our previous study and the increase is likely a

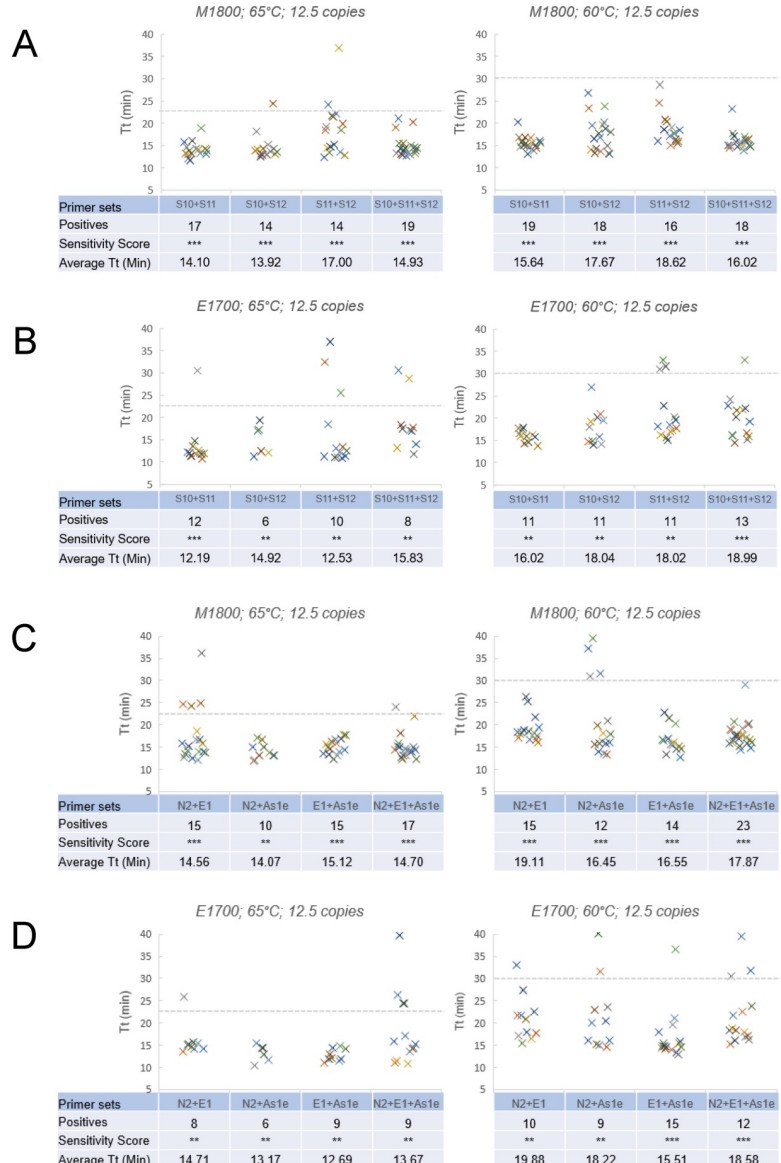

**Fig 4. Effect of combining 3 primer sets.** Detection sensitivity by a combination of 2 primer sets was compared with that by a combination of 3 primer sets in the presence of ~12.5 copies of template RNA per reaction in M1800 and E1700 at 65 and 60˚C. (A) Combinations of 2 primer sets for S10, S11 and S12 versus a combination of all 3 sets using M1800 at 65 and 60˚C. (B) S10, S11 and S12 in E1700 at 65 and 60˚C. (C) Combinations of 2 primer sets for N2, E1 and As1e versus a combination of all 3 sets using M1800 at 65 and 60˚C. (D) N2, E1 and As1e in E1700 at 65 and 60˚C. Sensitivity Score for 12.5 copies: *** 12–24 positives; ** 6–11 positives; * 1–5 positives.

result of combined probability of detection in the reaction [10]. All pairwise combinations of 2 primer sets showed increased detection sensitivity than any single primer set. A comparison of 4 sets of 2 primer combinations (S10+S11, S10+S12, S11+S12 and N2+E1) showed similar detection sensitivity in both M1800 and E1700 at both temperatures. We recommend these combinations and likely any combinations of these 5 sets (N2, E1, S10, S12 and S12) as most sensitive. We also tested combining 3 sets of primers into the same reaction. One group (N2 +E1+As1e) seemed to give further increased detection sensitivity as we described previously [10] but not with the other group (S10+S11+S12). The use of 2 sets together consistently

enhanced sensitivity and could be considered for any application where that is important, with the use of 3 primer sets a possible additional step but less likely to provide a further benefit. In addition to increasing sensitivity, combining primer sets for different gene targets reduces the potential impact of sequence mutations and variants that may arise in the targeted areas.

While the study presented here is limited to synthetic RNA control templates, the choice of optimal primer set and the sensitivity benefit of primer set combination is a fundamental starting point for any diagnostic assay. The benefit of combining RT-LAMP primer sets for increased sensitivity as we described in [10] has since been demonstrated with extracted RNA and directly from clinical samples [21–24] and thus we believe the analysis of many additional potential primer sets described here is of benefit to additional assay development. SARS-CoV-2 unfortunately remains a significant public health concern with continuing needs for diagnostic and surveillance testing, increasingly in field, point-of-care, and even home settings removed for traditional clinical laboratory infrastructure where RT-LAMP is particularly valuable. Coupled with proper validation of different sample sources and processing methods, these conditions and recommendations could significantly improve the diagnostic sensitivity of detecting SARS-CoV-2 and any future diagnostic targets with RT-LAMP.

## Supporting information

**S1 Table. Sequences of RT-LAMP primers.** All primers used in this study are shown here, with names as we refer to them and their original sources where appropriate. Amplicon size covers the F3-B3 distance as maps to the SARS-CoV-2 (MN908947.3), with positions on the genome listed.
(PDF)

## Acknowledgments

The authors are grateful to New England Biolabs for fostering an environment of scientific discussion and collaboration, without which this work would not have been possible.

## Author Contributions

**Conceptualization:** Yinhua Zhang, Nathan A. Tanner.

**Investigation:** Yinhua Zhang.

**Project administration:** Nathan A. Tanner.

**Supervision:** Nathan A. Tanner.

**Writing – original draft:** Yinhua Zhang, Nathan A. Tanner.

**Writing – review & editing:** Yinhua Zhang, Nathan A. Tanner.

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
