## [Decision Letter · Decision Letter 0]

17 Feb 2022

PONE-D-21-18467Improving RT-LAMP Detection of SARS-CoV-2 RNA through Primer Set Selection and CombinationPLOS ONE

Dear Dr. Tanner,

Thank you for submitting your manuscript to PLOS ONE. After careful consideration, we feel that it has merit but does not fully meet PLOS ONE’s publication criteria as it currently stands. Therefore, we invite you to submit a revised version of the manuscript that addresses the points raised during the review process.

Many apologies for the length of time that the review process has taken. In light of the referees comments, I would like to invite you to submit a revised manuscript. Please let me know if yourequire any additional time to resubmit and I am sure the Journal office can accommodate. Please submit your revised manuscript by Apr 03 2022 11:59PM. If you will need more time than this to complete your revisions, please reply to this message or contact the journal office at plosone@plos.org. Please include the following items when submitting your revised manuscript:A rebuttal letter that responds to each point raised by the academic editor and reviewer(s). You should upload this letter as a separate file labeled 'Response to Reviewers'.A marked-up copy of your manuscript that highlights changes made to the original version. You should upload this as a separate file labeled 'Revised Manuscript with Track Changes'.An unmarked version of your revised paper without tracked changes. You should upload this as a separate file labeled 'Manuscript'.

We look forward to receiving your revised manuscript.

Kind regards,

Eric C. Dykeman, Ph.D.

Academic Editor

PLOS ONE

Journal Requirements:

3. Thank you for stating the following in the Competing Interests section: "Authors are employed by New England Biolabs, manufacturer of reagents described in the manuscript."

We note that you received funding from a commercial source: New England Biolabs

5. Please include a copy of Table 1 which you refer to in your text on page 3.

Reviewers' comments:

Reviewer's Responses to Questions

**Comments to the Author**

1. Is the manuscript technically sound, and do the data support the conclusions?

Reviewer #1: Partly

Reviewer #2: Yes

2. Has the statistical analysis been performed appropriately and rigorously? 

Reviewer #1: Yes

Reviewer #2: I Don't Know

3. Have the authors made all data underlying the findings in their manuscript fully available?

Reviewer #1: Yes

Reviewer #2: Yes

4. Is the manuscript presented in an intelligible fashion and written in standard English?

Reviewer #1: Yes

Reviewer #2: Yes

5. Review Comments to the Author

Reviewer #1: In this work Zhang and Tanner evaluated SARS-CoV-2 primer sets for RT-LAMP assays in order to optimize test sensitivity. The topic is relevant because although RT-LAMP is easy to perform and allows high-speed amplification its use in SARS-CoV-2 diagnostics is limited because of a somewhat lower sensitivity compared to RT-PCR. The main result presented here is the definition of primer sets for a dual target assay with improved sensitivity.

Specific comments:

1. Material and methods, RT-LAMP reactions: Please specify the composition of the master mix. Which volume of RNA was pipetted into the PCR wells?

2. Results, procedure to identify sensitive primer sets: The authors state that they used 50, 25 and 12.5 virus copies: per microliter, ml or well?

3. Results, improving sensitivity … : The authors state that in all combinations tested the combinations gave higher sensitivity than with single primer sets. As shown in Fig. 2A this seems not to be the case for S12+S17, compared to S12 alone, right?

4. It seems to be better to use 60oC instead of 65oC for amplification (Fig. 2). Please specify.

5. Fig. 3: Why S12 and As1e have been chosen for reference when they were not tested in combination? These data should be omitted.

6. Fig 3: In my opinion at 12.5 virus copies (per well?) there is only one primer set (S10+S11 at 60oC) with convincing results of 20/24 positives (83.3%). Please specify.

7. It is unclear whether the assay should be used only for testing of extracted RNA but also for testing crude sample extracts without nucleic acid purification. To verify the findings obtained with a reference RNA for a practical usefulness of the assay, virus stock dilution series should be prepared in UTM/VTM medium and tested with or without RNA extraction.

Reviewer #2: This is an extremely important manuscript. The pandemics induces a huge spread of the LAMP technique and it is now used widely for various purposes. A comparison of primers for the detection of the virus is very important, since this cannot be done in silico and despite all the predictive algorhitms, it is still associated with a very high technical variability. Nevertheless, the manuscript itself requires corrections and improvements before publication. The abstract does not contain any information about the methods and the results. The reader does not know what has been tested and what is the outcome. The results should be at least partially quantitative.

Synthetic RNA is fine, but it is not the viral RNA found in biological samples, at least when it comes to the fragmentation status. This should at least be discussed. Also, synthetic RNA is pure... RNA from biological samples is mainly from the host, from bacteria and only a part of it is from the virus... again, this should at least be discussed, because it is not for sure that the sensitivity under these ideal conditions will be the same with real samples. The authors should also discuss the issues in real life applications - for example using saliva without RNA isolation, the robustness of the assay is likely more important than the sensitivity.

What was the reason for the cutoff of 22.5 minutes?

What is the opinion of the authors regarding the choice of the mastermix since pH dependency is a major issue for real life biological specimens, especially, for saliva. This can cause false positivity...

The authors do not really explain what could be the reason for the better sensitivity of some of the primer sets.

Why did the authors choose this design with 50 copies in every reaction? Why did they not compare the reactions with dilutions/calibration curves with potentially less replicates?

What is the reason for the occuring false negativity for some primer sets in some reactions?

If some of the primer combinations worked well with 12.5 copies, why did the authors no go even further down with the template copy number?

Average Tt... is it ok? should it not be median and quartiles of Tt in the reporting tables?

The conclusion is rather weak. The authors should clearly state the "winning team" and also suggest further research needs.

6. PLOS authors have the option to publish the peer review history of their article (what does this mean?). If published, this will include your full peer review and any attached files.

Reviewer #1: No

Reviewer #2: No

---

## [Author Response · Author response to Decision Letter 0]

4 Mar 2022

Reviewer Comments and Author Responses:

Reviewer #1: In this work Zhang and Tanner evaluated SARS-CoV-2 primer sets for RT-LAMP assays in order to optimize test sensitivity. The topic is relevant because although RT-LAMP is easy to perform and allows high-speed amplification its use in SARS-CoV-2 diagnostics is limited because of a somewhat lower sensitivity compared to RT-PCR. The main result presented here is the definition of primer sets for a dual target assay with improved sensitivity.

Specific comments:

1. Material and methods, RT-LAMP reactions: Please specify the composition of the master mix. Which volume of RNA was pipetted into the PCR wells? 

AUTHORS: We have added additional information on the reaction setup, as shown below. The master mix compositions are not publicly available, as is the case with every commercial PCR or LAMP master mix we know of, but we have shared all relevant information for the reactions as performed. Relevant section of the Methods now reads:

Primers were dissolved and then mixed in ddH2O as 25x stocks of each set based on standard 1x final concentrations in LAMP: 0.2 μM F3, 0.2 μM B3, 1.6 μM FIP, 1.6 μM BIP, 0.4 μM Loop F, 0.4 μM Loop B.

RT-LAMP reactions RT-LAMP reactions were performed using either WarmStart® Colorimetric LAMP 2X Master Mix (DNA & RNA) (M1800) or WarmStart® LAMP Kit (DNA & RNA) (E1700) containing Bst 2.0 WarmStart and WarmStart RTx polymerases from New England Biolabs (NEB). 40 mM guanidine hydrochloride was included in all reactions to improve LAMP reaction speed and sensitivity [10]. The same vial of synthetic SARS-CoV-2 RNA from Twist Bioscience (Twist Synthetic SARS-CoV-2 RNA Control 2 (MN908947.3), SKU: 102024) was used for all reactions, with SARS-CoV-2 RNA diluted and aliquoted in nuclease-free water and 10 ng/μl Jurkat total RNA (Biochain). RT-LAMP reactions were performed in 25 μl volumes with 1 ul of diluted SARS-CoV-2 RNA supplemented with 1 μM SYTO®-9 double-stranded DNA binding dye (Thermo Fisher S34854) in 96-well plates and incubated at 65 or 60 °C on a real-time qPCR machine (BioRad CFX96). 

2. Results, procedure to identify sensitive primer sets: The authors state that they used 50, 25 and 12.5 virus copies: per microliter, ml or well? 

AUTHORS: We thank the reviewer for pointing out this point of uncertainty. We have updated the reaction description to now read: “…using 50 copies of SARS-CoV-2 synthetic RNA in each 25 ul reaction.” Additionally we have added “per reaction” to each figure legend where the copy number is described. 

3. Results, improving sensitivity … : The authors state that in all combinations tested the combinations gave higher sensitivity than with single primer sets. As shown in Fig. 2A this seems not to be the case for S12+S17, compared to S12 alone, right? 

AUTHORS: The reviewer’s analysis of this data is correct, and we appreciate the opportunity to clarify this point. S12+S17 did not show improved sensitivity at both 65 °C and 60 °C, only at 60 °C. The relevant section now reads: 

In all the combinations tested (S10+S11; S10+S12; S11+S12; and N2+E1), the combined primer reactions gave higher sensitivity than with single primer sets at both temperatures, except S12+S17 which matched the performance of S12 at 65 °C but increased sensitivity at 60 °C.

4. It seems to be better to use 60oC instead of 65oC for amplification (Fig. 2). Please specify. 

AUTHORS: In several cases use of 60 °C does indeed result in improved sensitivity. We had described this in the original version but have updated the relevant text to now read:

(RESULTS) The results showed that there were slight differences in their performance regarding formulation and incubation temperature. N2+E1 showed slightly higher sensitivity at 60°C in both formulations while combinations of S10, S11, and S12 showed the same tendency only in E1700. 

… As shown, combination of 2 primer sets S10+S11 (Figure 3A-B) was clearly more sensitive with M1800 at 60°C (20/24 positives vs 13/24 and 8/24 single sets) but less pronounced at 65°C (13/24 positives vs 8/24 and 6/24 single sets). A similar trend was observed with E1700 but with a lower positive frequency in all cases than with M1800. N2+E1 (Figure 3C-D) also showed more sensitive detection than each single primer set with M1800 at both 65°C (16/24 positives vs 9/24 and 6/24 single sets) and 60°C (18/24 positives vs 10/24 and 9/24 single sets) and with E1700 at 60 °C (12/24 positives vs 4/24 and 5/24 single sets) but less clear at 65°C (7/24 positives vs 6/24 and 3/24 single sets). These results reinforced the concept that RT-LAMP with dual primer set is more sensitive even at low template and for some primer sets use of 60 °C can provide increased sensitivity albeit with potentially increased reaction times.

(DISCUSSION) Overall we did observe some consistent trends regarding to reaction formulation and temperature, with a slight improvement in sensitivity with the M1800 Colorimetric LAMP as compared to the general-purpose E1700. Optimal reaction temperature varied for the 18 sets. Using M1800, 13 sets showed similar performance at the two temperatures, 3 more sensitive at 60 °C and 2 more sensitive at 65 °C. Using E1700 there was more variation, with 6 sets giving similar results, 7 more sensitive at 60 °C and 5 more sensitive at 65 °C. In general, primers performed well and primers that are poor remains the same in both recipes and reaction temperatures, suggesting that the identification of good RT-LAMP primer is the most important step toward sensitive and reliable assay. 

5. Fig. 3: Why S12 and As1e have been chosen for reference when they were not tested in combination? These data should be omitted. 

AUTHORS: We have removed the data for S12 and As1e for clarity in Figure 3. 

6. Fig 3: In my opinion at 12.5 virus copies (per well?) there is only one primer set (S10+S11 at 60oC) with convincing results of 20/24 positives (83.3%). Please specify. 

AUTHORS: We agree that at this lowest copy number per reaction (now clarified) few of the primer combinations provide sufficient sensitivity for assigning an assay limit of detection in this range. But we are not doing so, merely using low template concentrations where the standard reaction typically fails to demonstrate that primer combination increases the likelihood of detection. We have updated this section to be more clear, with the text now reading:

We next extended this comparison with even lower SARS-CoV-2 RNA templates (~12.5 copies) to test how these primers perform as single or dual primer sets (Figure 3). 4 primer sets were divided into two groups (S10, S11; and N2, E1) and the single primer set was compared with dual primer set from each group (S10+S11, N2+E1). As shown, combination of 2 primer sets S10+S11 (Figure 3A-B) was clearly more sensitive with M1800 at 60°C (20/24 positives vs 13/24 and 8/24 single sets) but less pronounced at 65°C (13/24 positives vs 8/24 and 6/24 single sets). A similar trend was observed with E1700 but with a lower positive frequency in all cases than with M1800. N2+E1 (Figure 3C-D) also showed more sensitive detection than each single primer set with M1800 at both 65°C (16/24 positives vs 9/24 and 6/24 single sets) and 60°C (18/24 positives vs10/24 and 9/24 single sets) and with E1700 at 60 °C (12/24 positives vs 4/24 and 5/24 single sets) but less clear at 65°C (7/24 positives vs 6/24 and 3/24 single sets). These results reinforced the concept that RT-LAMP with dual primer set is more sensitive even at low template and for some primer sets use of 60 °C can provide increased sensitivity.

Combination of 3 primer sets. As use of 2 primer sets showed better detection sensitivity than a single set, we evaluated whether adding a third set could further improve the detection sensitivity. We assayed two groups (S10, S11, S12; and N2, E1, As1e) with all pairwise combinations of primer sets within each group versus reactions containing all 3 sets. LAMP reactions were performed with ~12.5 copies of SARS-CoV-2 RNA template. Overall, the benefit of 3 primer sets over 2 sets was not consistent across different primer combinations. For S10+S11+S12 (Figure 4A-B), there seemed to be no advantage and it gave similar number of positives as those reactions with only 2 sets of primers combinations (S10+S11, S10+S12 and S11+S2) in both RT-LAMP reagents and at either reaction temperature. For N2+E1+As1e (Figure 4C-D), there was some advantage, and it produced more positives than those reactions containing 2 primer sets (N2+E1, N2+As1e and E1+As1e) with M1800 especially at 60 °C but not with E1700. While of course it is possible that other combinations of 3 primer sets may provide a sensitivity benefit to RT-LAMP, the results seen here indicate diminishing returns adding a 3rd set to a combination of 2 primer sets, and that a combination of 2 sets is likely a sufficient approach to increasing sensitivity. 

7. It is unclear whether the assay should be used only for testing of extracted RNA but also for testing crude sample extracts without nucleic acid purification. To verify the findings obtained with a reference RNA for a practical usefulness of the assay, virus stock dilution series should be prepared in UTM/VTM medium and tested with or without RNA extraction. 

AUTHORS: We do not present our study as a validated clinical assay but merely evaluating the performance of RT-LAMP with the many described primer sets. RT-LAMP assays for SARS-CoV-2 have been published in great numbers, with dozens approved for diagnostic use around the world so we do not feel that is a point we need to prove here. Our RNA samples are all diluted in a background of Jurkat total RNA and match what would be present in RNA extracted from real samples. And numerous studies have taken our original limited primer combination study and demonstrated improved sensitivity with real samples, and we have added references to several of those in the updated discussion of this point, amended text now reads: 

While the study presented here is limited to synthetic RNA control templates, the choice of optimal primer set and the sensitivity benefit of primer set combination is a fundamental starting point for any diagnostic assay. The benefit of combining RT-LAMP primer sets for increased sensitivity as we described in [10] has since been demonstrated with extracted RNA and directly from clinical samples [21-24] and thus we believe the analysis of many additional potential primer sets described here is of benefit to additional assay development. SARS-CoV-2 unfortunately remains a significant public health concern with continuing needs for diagnostic and surveillance testing, increasingly in field, point-of-care, and even home settings removed for traditional clinical laboratory infrastructure where RT-LAMP is particularly valuable. Coupled with proper validation of different sample sources and processing methods, these conditions and recommendations could significantly improve the diagnostic sensitivity of detecting SARS-CoV-2 and any future diagnostic targets with RT-LAMP.

Reviewer #2: This is an extremely important manuscript. The pandemics induces a huge spread of the LAMP technique and it is now used widely for various purposes. A comparison of primers for the detection of the virus is very important, since this cannot be done in silico and despite all the predictive algorhitms, it is still associated with a very high technical variability. Nevertheless, the manuscript itself requires corrections and improvements before publication. The abstract does not contain any information about the methods and the results. The reader does not know what has been tested and what is the outcome. The results should be at least partially quantitative. 

AUTHORS: We have updated the Abstract to contain additional information about the methods and results described in the study. 

Synthetic RNA is fine, but it is not the viral RNA found in biological samples, at least when it comes to the fragmentation status. This should at least be discussed. Also, synthetic RNA is pure... RNA from biological samples is mainly from the host, from bacteria and only a part of it is from the virus... again, this should at least be discussed, because it is not for sure that the sensitivity under these ideal conditions will be the same with real samples. The authors should also discuss the issues in real life applications - for example using saliva without RNA isolation, the robustness of the assay is likely more important than the sensitivity. 

AUTHORS: This is a similar point raised by Reviewer #1, and we feel the updated discussion on this point covers this. Again, we dilute all our RNA in a background of Jurkat total RNA to mimic real samples. Performance of RT-LAMP with saliva or other direct samples is beyond the scope of this study but has been demonstrated by numerous others with many of the primer sets we describe, and additional references in the section helps address the point. 

What was the reason for the cutoff of 22.5 minutes? Added in the method “The cutoff times are somewhat arbitrary but reflect that efficient LAMP amplifications occur within the cutoff.”

AUTHORS: The reviewer is correct that the cutoff time is arbitrary, but any such time would be as well and we compare the primer sets and conditions against each other with the same cutoff and all performance is relative to that. We updated a sentence to the Methods section stating as much: “The time to reach the signal threshold (Tt) was determined from the real time fluorescence signal and positive was scored using an arbitrary cutoff of 22.5 minutes for RT-LAMP at 65 °C and 30 minutes at 60 °C in order to compare LAMP efficiency.”

What is the opinion of the authors regarding the choice of the mastermix since pH dependency is a major issue for real life biological specimens, especially, for saliva. This can cause false positivity.

AUTHORS: Our opinion is that this point is beyond the scope of the current study and best described in the many others we reference looking at LAMP or pH-based colorimetric LAMP with different sample types. These points are very relevant but depend entirely on the sample of choice, and simple solutions like use of an alkaline lysis buffer in several of the references make saliva perfectly compatible with the colorimetric LAMP mix. Statements toward the end of the discussion as described above sufficiently address this point. 

The authors do not really explain what could be the reason for the better sensitivity of some of the primer sets. 

AUTHORS: We do not explain the reason for this because we do not know it. It would be great if we could explain or describe how to know a certain primer set will give highest sensitivity but unfortunately empirical testing is needed to determine performance, hence the study here and others like it. But to further discuss to this point we added text to the Discussion with a reference to one thermodynamic approach to LAMP primer optimization (albeit one that does not claim to enhance sensitivity). 

The ability to design LAMP primers in provided by LAMP primer tools, but empirically testing each designed set for performance remains the best strategy to identify highest performance. Adjustments to designed primers such described by [20] can improve curve shape and specificity, but for maximum sensitivity screening and combination are the best suggestions for assay development. 

Why did the authors choose this design with 50 copies in every reaction? Why did they not compare the reactions with dilutions/calibration curves with potentially less replicates? 

AUTHORS: We chose 50 copies as it approximates the limit of detection of many good RT-LAMP assays, and comparing performance at that level allows for distinction of performance between high- and low-sensitivity primer sets. Comparing at high copy numbers would only give a sense of reaction speed, which does not vary widely with these (all pretty good) primer sets. We do look at lower copy numbers, with many replicates at 25 or 12.5 copies to more confidently compare the performance, and we are mimicking limit of detection measurements by establishing detection frequencies at low inputs. 

What is the reason for the occurring false negativity for some primer sets in some reactions? 

AUTHORS: As above, we don’t know the answer to this question. Some thermodynamic factors of the primers is likely responsible but it is more complex than any obvious parameter/design feature can account for. Again, empirical screening of any potential RT-LAMP primer set is the best approach to identify the best performance as we describe and recommend. 

If some of the primer combinations worked well with 12.5 copies, why did the authors no go even further down with the template copy number? 

AUTHORS: As shown in the data and as pointed out by Reviewer 1 the RT-LAMP performance even with the best primer sets clearly decreased at this low copy input condition. Differences are very apparent with the results as they are and we don’t see any benefit to decreasing the input, where stochasticity from sample input would become an even greater factor and interfere with the increasingly variable results. 

Average Tt... is it ok? should it not be median and quartiles of Tt in the reporting tables?

AUTHORS: We feel the average Tt is a sufficient metric for comparing the different primer sets, and the outliers beyond the cutoff zone are already excluded from the analysis to not skew the average. Grouping tends to be very consistent for one primer set or one combination, so the median is unlikely to be different than the average, nor will the quartiles add much beyond complicating already very data-dense figures. 

The conclusion is rather weak. The authors should clearly state the "winning team" and also suggest further research needs.

AUTHORS: The reviewer is correct, and we had written it to not strongly pick a certain condition or primer set. But we have update this to describe the best primer sets and conditions, see below, and have added additional discussion of the validation on real samples as described above. 

A comparison of 4 sets of 2 primer combinations (S10+S11, S10+S12, S11+S12 and N2+E1) showed similar detection sensitivity in both M1800 and E1700 at both temperatures. We recommend these combinations and likely any combinations of these 5 sets (N2, E1, S10, S12 and S12) as most sensitive. We also tested combining 3 sets of primers into the same reaction. One group (N2+E1+As1e) seemed to give further increased detection sensitivity as we described previously [10] but not with the other group (S10+S11+S12). The use of 2 sets together consistently enhanced sensitivity and could be considered for any application where that is important, with the use of 3 primer sets a possible additional step but less likely to provide a further benefit.

---

## [Decision Letter · Decision Letter 1]

22 Mar 2022

Improving RT-LAMP Detection of SARS-CoV-2 RNA through Primer Set Selection and Combination

PONE-D-21-18467R1

Dear Dr. Tanner,

We’re pleased to inform you that your manuscript has been judged scientifically suitable for publication and will be formally accepted for publication once it meets all outstanding technical requirements.

Kind regards,

Eric C. Dykeman, Ph.D.

Academic Editor

PLOS ONE

Additional Editor Comments (optional):

If you could please note that one reviewer has asked for the sensitivity of the assay to be highlighted in the abstract. I've left this as an optional revision.

Reviewers' comments:

Reviewer's Responses to Questions

**Comments to the Author**

1. If the authors have adequately addressed your comments raised in a previous round of review and you feel that this manuscript is now acceptable for publication, you may indicate that here to bypass the “Comments to the Author” section, enter your conflict of interest statement in the “Confidential to Editor” section, and submit your "Accept" recommendation.

Reviewer #1: All comments have been addressed

Reviewer #2: All comments have been addressed

2. Is the manuscript technically sound, and do the data support the conclusions?

Reviewer #1: Yes

Reviewer #2: Yes

3. Has the statistical analysis been performed appropriately and rigorously? 

Reviewer #1: Yes

Reviewer #2: Yes

4. Have the authors made all data underlying the findings in their manuscript fully available?

Reviewer #1: Yes

Reviewer #2: Yes

5. Is the manuscript presented in an intelligible fashion and written in standard English?

Reviewer #1: Yes

Reviewer #2: Yes

6. Review Comments to the Author

Reviewer #1: (No Response)

Reviewer #2: The authors addressed all the issues mentioned. I still would like to see some quantitative results, at least the sensitivity in the abstract, but that is a minor issue.

7. PLOS authors have the option to publish the peer review history of their article (what does this mean?). If published, this will include your full peer review and any attached files.

Reviewer #1: No

Reviewer #2: No

---

## [Editor Report · Acceptance letter]

25 Mar 2022

PONE-D-21-18467R1 

Improving RT-LAMP Detection of SARS-CoV-2 RNA through Primer Set Selection and Combination 

Dear Dr. Tanner:

I'm pleased to inform you that your manuscript has been deemed suitable for publication in PLOS ONE. Congratulations! Your manuscript is now with our production department. 

Kind regards, 

on behalf of

Dr. Eric C. Dykeman 

Academic Editor

PLOS ONE